# The Effect of Water Transfer during Non-growing Season on the Wetland Ecosystem via Surface and Groundwater Interactions in Arid Northwestern China

**Shufeng Qiao, Rui Ma \*, Ziyong Sun, Mengyan Ge, Jianwei Bu, Junyou Wang, Zheng Wang and Han Nie**

School of Environmental Studies and MOE Key Laboratory of Biogeology and Environmental Geology, China University of Geosciences, Wuhan 430074, China; sqiao@cug.edu.cn (S.Q.); ziyong.sun@cug.edu.cn (Z.S.); myge@cug.edu.cn (M.G.); jwbu@cug.edu.cn (J.B.); jywang@cug.edu.cn (J.W.); wangzheng@cug.edu.cn (Z.W.); niehan1997@cug.edu.cn (H.N.)

**\*** Correspondence: rma@cug.edu.cn

**Abstract:** The use of ecological water transfer to maintain the ecological environment in arid or semiarid regions has become an important means of human intervention to alleviate vegetation ecosystem degradation in arid and semiarid areas. The water transfer to downstream in a catchment is often carried out during the non-growing season, due to the competitive water use between the upper and middle reaches and lower reaches of rivers. However, the impacts and mechanism of artificial water transfer on vegetation and wetland ecosystem restoration have not been thoroughly investigated, especially in northwest China. Taking the Qingtu Lake wetland system in the lower reaches of the Shiyang River Catchment as the study area, this study analyzed the spatial and temporal distribution surface area of Qingtu Lake and the surrounding vegetation coverage before and after water transfer, by interpreting remote sensing data, the variation of water content in the vadose zone, and the groundwater level by obtaining field monitoring data, as well as the correlation between the water body area of Qingtu Lake and the highest vegetation coverage area in the following year. The conclusion is that there is a positive correlation between the water body area of Qingtu Lake in autumn and the vegetation coverage in each fractional vegetation coverage (FVC) interval in the next summer, especially in terms of the FVC of 30–50%. The groundwater level and soil water content increase after water transfer and remain relatively high for the following months, which suggests that transferred water from upstream can be stored as groundwater or soil water in the subsurface through surface water and subsurface water interaction. These water sources can provide water for the vegetation growth the next spring, or support plants in the summer.

**Keywords:** ecological water transfer; wetland vegetation ecosystem; surface and groundwater interaction; northwestern China; remote sensing

## 1. Introduction

The arid and semiarid regions in the world usually have fragile ecological environments due to low precipitation and a lack of water resources. Excessive use of water resources leads to ecological system degradation, including wetland degradation and plant ecosystem deterioration [1,2]. For example, climate change and water resource exploitation currently threaten groundwater-dependent ecosystems and put vegetation at risk of degradation in the Nalenggele River Basin, located in the southwest Qaidam Basin in the Qinghai province of northwest China [3]. Due to intense water usage in the

Junggar Basin of northwestern China over the last few decades, the water flowing into Ebinur Lake has been greatly reduced. Thus, Ebinur Lake, which lies on the southwest margin of the basin, has been continuously shrinking; the water area had shrunk from over 1000 km$^2$ to less than 500 km$^2$ in 2011. The water reduction of Ebinur Lake has led to a serious recession of lakeshore vegetation, with 60% of the desert forest around the lake vanishing, and the desertification area is expanding at a speed of 39.8 km$^2$ per year on average [4]. Similar situations have also occurred in other countries, such as Australia [5], Mexico [6,7], Nepal [8], the North Basin in Kenya [9], Iran, and Afghanistan [10]. In the Narran Lakes in Australia, one of the most important Ramsar-listed wetlands due to its provision of habitat for wetland fauna during key life history stages, reduced ibis breeding due to water resource exploitation has been reported, from 1 in 4.2 years to 1 in 11.4 years [5]. The increase of water withdrawal and introduction of exotic herbage species have aggravated the transformation of the ecosystem in San Miguel and Zanjon River in the northwest of Mexico, such that the range of crops and grasslands have been significantly reduced and desert shrub species significantly increased [7]. The excessive abstraction of Ewaso Ng'iro River water in the Upper Ewaso Ng'iro North Basin in Kenya has greatly affected downstream water users, and led to the deterioration of the vegetative cover and a reduction in water flow in the Ewaso Ng'iro River and its major tributaries [9]. Hence, the excessive use of water resources can lead to a decline of the groundwater level. Problems associated with this include salinization, land subsidence, and deterioration of water quality, which could be harmful to vegetation and cause ecosystem degradation [11].

Given the current situation, the restoration of wetlands has become an important option for the international community. However, ecological reversal to the natural status is a long-term dynamic change process [12]. People are working to accelerate the restoration of the ecosystem through manual intervention, and ecological water transportation to ecologically fragile areas is one of the most important methods to achieve this [13,14]. In northwest China, this measure has been employed to maintain the ecological system balance of downstream waters, especially for wetland vegetation systems. For example, Xinjiang's Tarim River received water from a water transfer project during ecological emergency periods eight times between 2001 and 2006; the transfer improved groundwater quality and restored natural grasslands [15]. Similarly, a total of 4.512 billion m$^3$ of water was transferred to the downstream area in the Heihe River Basin from 2000 to 2008, which led to extensive vegetation restoration [16]. Maintaining the ecosystem in downstream waters through ecological water transportation has become the focus of ecological research of arid areas throughout the world [13,14]. Due to the demand for water resources in the upper and middle reaches of these arid river basins in spring and summer for agriculture, little water tends to be available to transfer downstream. Normally, waters in autumn, when the water demand is alleviated in the upstream, are transferred downstream. However, vegetation normally stops growing during this period. Will the water transferred in autumn help maintain the vegetation ecosystem in spring and summer, and if so, how? The quantity and timing of water transfer can determine the effect of water transfer on ecological restoration. However, few studies have been reported on this issue.

Shiyang River is China's third-largest inland river. Its lower reaches are located in the Minqin Basin, the wedge of which is at the cross-point of the Badain Jaran Desert and Tengger Desert. This oasis is an important barrier to prevent inland movement of big sandstorms from the desert. Qingtu Lake, located at the northern edge of Minqin Basin, is the terminal lake of the Shiyang River. It has important ecological significance in preventing the connection of the two big deserts [17]. Because of the construction of a water conservancy project in the upper reaches, and the Hongyashan Reservoir in the middle reaches of Shiyang River, the main channel of Shiyang River in the lower reaches had been dry for a long time, which led to Qingtu Lake, the terminal lake of the Shiyang River, drying in 1959. As a result, the wetland ecosystem and vegetation was transformed into desert, and most areas near the wetland were covered by quicksand, while the regional ecosystem continued to deteriorate [18]. In order to improve the ecological environment in the lower reaches of Shiyang River, the government promoted a water transfer project that transports ecological water into Qingtu Lake downstream via

a channel. This started in September 2010 and has continued in autumn every year since. By November 2016, a 25.16 km$^2$ area of water body surface had formed in the Qingtu Lake [19]. Due to conflict around the ecological water demand in the lower reaches and the agricultural water needs in the middle and upper reaches of Shiyang River in spring and summer, ecological water transportation has been implemented in autumn, when the vegetation has almost stopped growing. However, the impacts and mechanism of artificial water transfer on vegetation and wetland ecosystem restoration remain poorly understood. Taking the Qingtu Lake wetland ecosystem in the lower reaches of the Shiyang River Basin as an example, this study investigated the effect of the water transfer in autumn on the wetland vegetation ecosystem, and explored the mechanism of how water input during the non-growing season improves the vegetation ecosystem.

## 2. Study Area

Qingtu Lake, the terminal lake of the Shiyang River, is located on the northeast edge of Minqin County, Gansu Province, between 39°04′ and 39°09′ N latitude and 103°36′ and 103°39′ E longitude (Figure 1). The area has a temperate continental arid desert climate. The mean annual temperature is 8.87 °C; the maximum temperature is 37.8 °C and the minimum temperature is −29.5 °C. The mean annual precipitation is 110 mm, with 73% of this occurring in June, July, and August. The mean annual evaporation is 2644 mm [20]. The main vegetation species include *Phragmites australis*, *Nitraria tangutorum*, *Suaeda glauca*, *Haloxylon ammodendron*, and *Kalidium foliatum*. Accompanying shrubs include *Lycium ruthenicum*, *Artemisia sphaerocephala*, and *Kalidium foliatum*. Herbaceous plants include those such as *Salsola ruthenica*, *Zygophyllum fabago*, *Cynanchum sibiricum*, *Salsola collina*, and *Agriophyllum squarrosum*. They are all typical desert vegetation types [21]. The water of Qingtu Lake is mainly used to maintain the ecological system balance around the wetland, especially for improving the vegetation ecosystem [20,22]. The water in Qingtu Lake is weakly alkaline and Na–SO$_4$ (Cl) type, with high salinity and total dissolved solids (TDS) [23]. The lithology is shown in the cross section (Figure 2). The sediments in this area consist of mainly sand, silt, and loam [24].

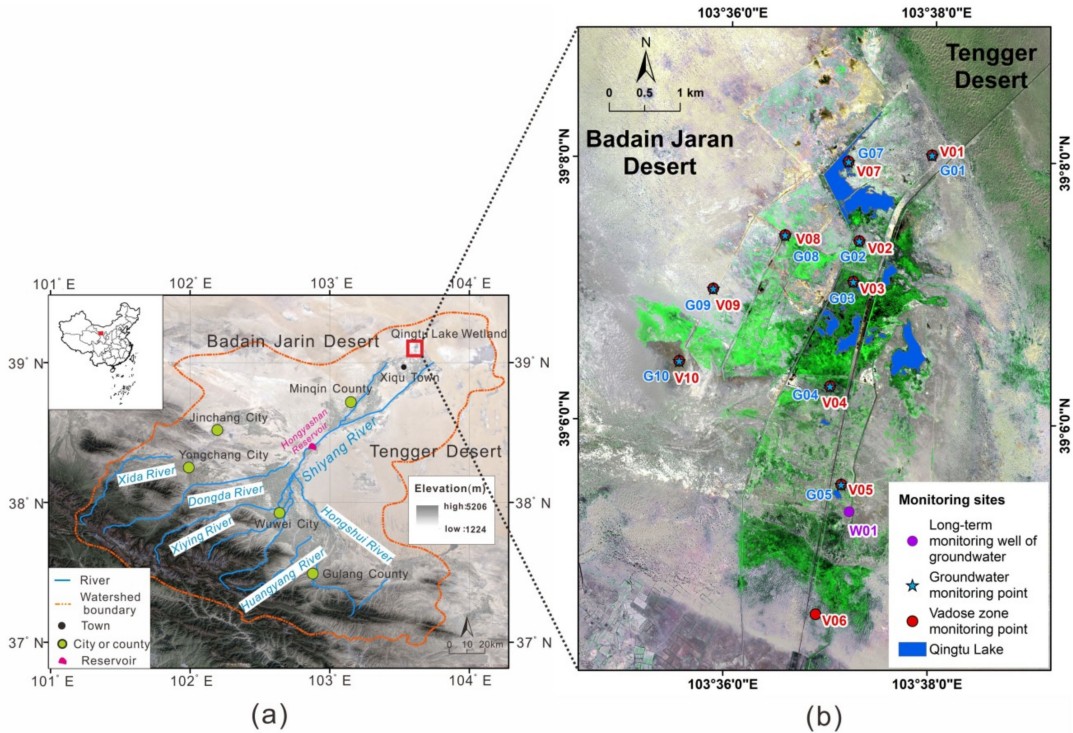

**Figure 1.** Location of the study area (**a**) and monitoring locations in the Qingtu Lake wetland (**b**).

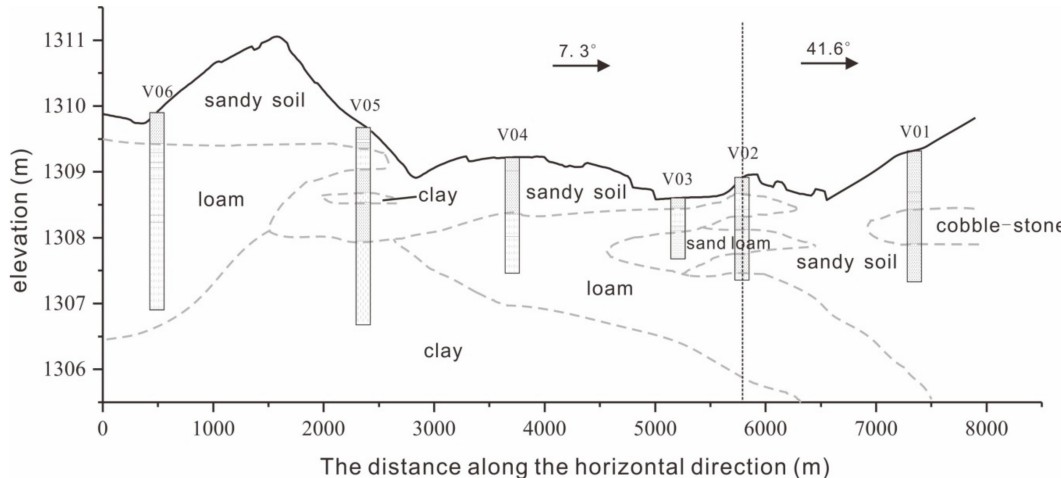

**Figure 2.** Simplified geological cross section from V01 to V06 (locations are shown in Figure 1b).

During the Western Han Dynasty, Qingtu Lake covered an area of 4000 km$^2$. Due to the increase of water consumption in the upper reaches of Qingtu Lake, the surface area of the water body of Qingtu Lake began to gradually diminish after 1924. In the 1950s, the construction of the Hongyashan Reservoir caused the Shiyang River to dry quickly [25]. Since the 1960s, there have been a series of ecological problems in the lower reaches of Shiyang River, such as the decline of the groundwater level, vegetation degradation, and soil salinization [23]. In order to restore the regional ecological environment, the local government decided to begin transferring ecological water from Hongyashan Reservoir to Qingtu Lake in September 2010. The water is transferred through irrigation channels from September to October of each year. It is a typical ecological water transportation system, carried out during the non-growing season.

## 3. Data and Methods

### 3.1. Data Sources and Preprocessing

Ten monitoring sites were constructed in July and August 2018. Instrumentation at the V01–V10 locations was implemented using 5TE probes developed by the Decagon Company, with a precision of 3% for monitoring the soil water content at different depths along the soil profile (Figure 1b). The G01–G10 wells with screens at depths between 1 and 3.6 m were installed with Canadian Solinst3001 titanium gold edge water-level loggers with a precision of 0.05% for recording the groundwater level, water salinity, and temperature (Figure 1b). The soil water content and parameters in the groundwater were recorded every 30 min. Meteorological data in the study area (temperature, precipitation, air pressure) were downloaded from the China Meteorological Data Sharing Network (http://data.cma.cn).

Considering the timing of water transfer from Hongyashan Reservoir to Qingtu Lake and vegetation evolution, the remote sensing data used in this study were acquired from Landsat series satellite data, including Landsat 8 OLI, Landsat 7 OFF, and Landsat 4-5 TM from 2009 to 2018, with a resolution of 30 m, a synthetic image resolution of 15 m, and a temporal resolution of 16 d. The remote sensing data from after May 2013 were from the Landsat 8 OLI series, while data from before May 2013 were from the Landsat 7 OFF or Landsat 4-5 TM series. The data from the Landsat 7 OFF series were damaged; the images showed data overlap and about 25% of the data was lost. All of the satellite data of the Landsat 7 OFF series used in this study were repaired by the strip removal method [26]. The available Landsat data from before May 2013 were limited, and there were nine images from 2009 to 2018.

All the image data were preprocessed with ENVI 5.2 for radiometric calibration, layer stacking, atmospheric correction, geometric correction, and cropping [27].

*3.2. Methods*

3.2.1. Water Extraction Methods

The NDWI (normalized difference water index) proposed by McFeeters was initially designed to enhance the difference between water and nonwater bodies, in order to generate initial water body maps. However, it cannot efficiently suppress the signal noise coming from the land cover features of built-up areas [28,29]. Hence, Xu [30] proposed a modified NDWI (MNDWI), which is defined as:

$$\text{MNDWI} = \frac{(\text{Green} - \text{MIR})}{(\text{Green} + \text{MIR})} \tag{1}$$

where Green is the green band, which corresponds to the TM/ETM + image band 2 (0.52–0.60 μm) and the OLI image band 3 (0.525–0.600 μm). MIR represents the mid-infrared band, which corresponds to the TM/ETM + image band 5 (1.55–1.75 μm) and OLI image band 6 (1.560–1.660 μm) [30,31].

Water bodies have positive values in the MNDWI, while soil, vegetation, and built-up classes tend to have negative values. Hence, the normal empirical threshold is zero [29].

3.2.2. Method for FVC Estimation

FVC (fractional vegetation cover) is defined as the fraction of green vegetation seen from nadir, which can characterize the growth conditions and horizontal density of land surface live vegetation. There are three major algorithms for FVC estimation, which include empirical methods, pixel unmixing models, and machine learning methods [32]. The dimidiate pixel model, which was used in this study, is a method of pixel unmixing and widely used for FVC estimation [33]. It is assumes the NDVI (normalized difference vegetation index) in a pixel consists of soil and vegetation, and the NDVI can be derived by [34,35]:

$$\text{NDVI} = \text{FVC} \times \text{NDVI}_{\text{veg}} + (1 - \text{FVC}) \times \text{NDVI}_{\text{soil}} \tag{2}$$

where $\text{NDVI}_{\text{veg}}$ represents the NDVI value of a pure vegetation pixel and $\text{NDVI}_{\text{soil}}$ represents the NDVI value of a pure soil pixel. The $\text{NDVI}_{\text{soil}}$ selected an NDVI value with a cumulative frequency of 0.5%, and the $\text{NDVI}_{\text{veg}}$ selected an NDVI value with a cumulative frequency of 99.5% [36,37].

Hence, the FVC can be derived by modifying Equation (2) as:

$$\text{FVC} = \frac{(\text{NDVI} - \text{NDVI}_{\text{soil}})}{(\text{NDVI}_{\text{veg}} - \text{NDVI}_{\text{soil}})} \tag{3}$$

3.2.3. Classification Method of FVC

For the entire Minqin Basin, there are few vegetation areas and vegetation types. The vegetation coverage types of the Minqin Basin can be divided into five main vegetation coverage categories [38]: extremely low, low, medium, medium and high, and high (Table 1).

**Table 1.** Vegetation Coverage Classification Standards [38].

| Class | Vegetation Coverage | Status of Fractional Vegetation Coverage (FVC) | Description |
|:---:|:---:|:---:|:---:|
| 1 | extremely low | FVC ≤ 10% | Intense desertified land, bare rock, bare soil, and waters |
| 2 | low | 10% < FVC ≤ 30% | Moderately desertified land, low yield grassland, and sparse woodland |
| 3 | medium | 30% < FVC ≤ 50% | Slightly desertified land, medium grassland, low canopy woodland, and cultivated land |
| 4 | medium and high | 50% < FVC ≤ 70% | Medium and high yield grassland, forest, and cultivated land |
| 5 | high vegetation | FVC > 70% | High yield grassland, forest, and cultivated land |

### 3.2.4. Correlation Analysis

This analysis aims to explore the influence of the former expansion of Qingtu Lake on the secondary vegetation coverage around Qingtu Lake after ecological water transfer. The two variables are one-to-one, corresponding, and relatively continuous, so the Pearson correlation coefficient was selected for calculation. The equation is given below [39].

$$r = \frac{\sum (X - \overline{X})(Y - \overline{Y})}{\sqrt{\sum (X - \overline{X})^2 \sum (Y - \overline{Y})^2}} \tag{4}$$

where X and Y represent the two groups of variables, and $\overline{X}$ and $\overline{Y}$ represent the average values of the two groups of variables. If r is greater than 0, then a positive correlation between the two variables is indicated; if the contrary, there is a negative correlation. In addition, the greater the absolute value of r, the stronger the correlation between the two variables.

## 4. Result

### 4.1. Spatial and Temporal Variations of the Water Body Area of Qingtu Lake from 2009 to 2018

#### 4.1.1. Temporal Change

Since September 2010, the ecological waters have been transferred from the reservoir to Qingtu Lake from September to October each year [40]. The surface area of the water body of Qingtu Lake exhibited a dynamic change trend (Figure 3). The lake water body reached the largest surface area from November to the following January, with a maximum area of 13.7 km$^2$. It normally exhibits the lowest area from July to August each year.

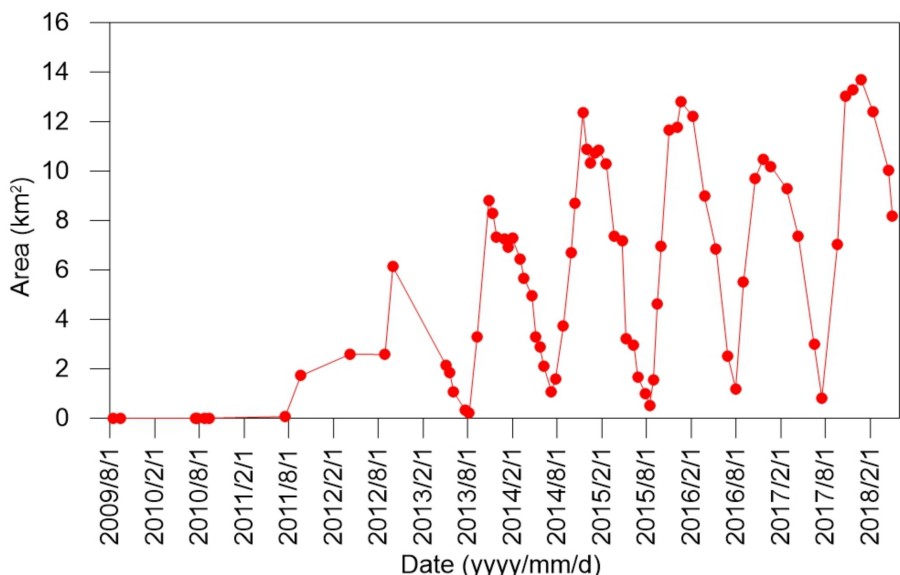

**Figure 3.** The change in the surface area of the water body of Qingtu Lake over time.

#### 4.1.2. Spatial Change

Due to the similar temporal trend of the Qingtu Lake water surface area each year, only the spatial distributions of Qingtu Lake water area in different months of 2017 are shown in Figure 4. The water surface area of Qingtu Lake changed little between January and February, ranging between 9.2 km$^2$ and 9.6 km$^2$ (Figure 4). After April, the water area began to shrink rapidly, and then reached the lowest level (0.8226 km$^2$) in July. From September, the water area began to expand again, since the water transfer to Qingtu Lake was started in September. The water body expanded to the south and north,

and then gradually spread to the west, reaching the highest lake level in November and December. The lake water level then almost stabilized from November to the following February.

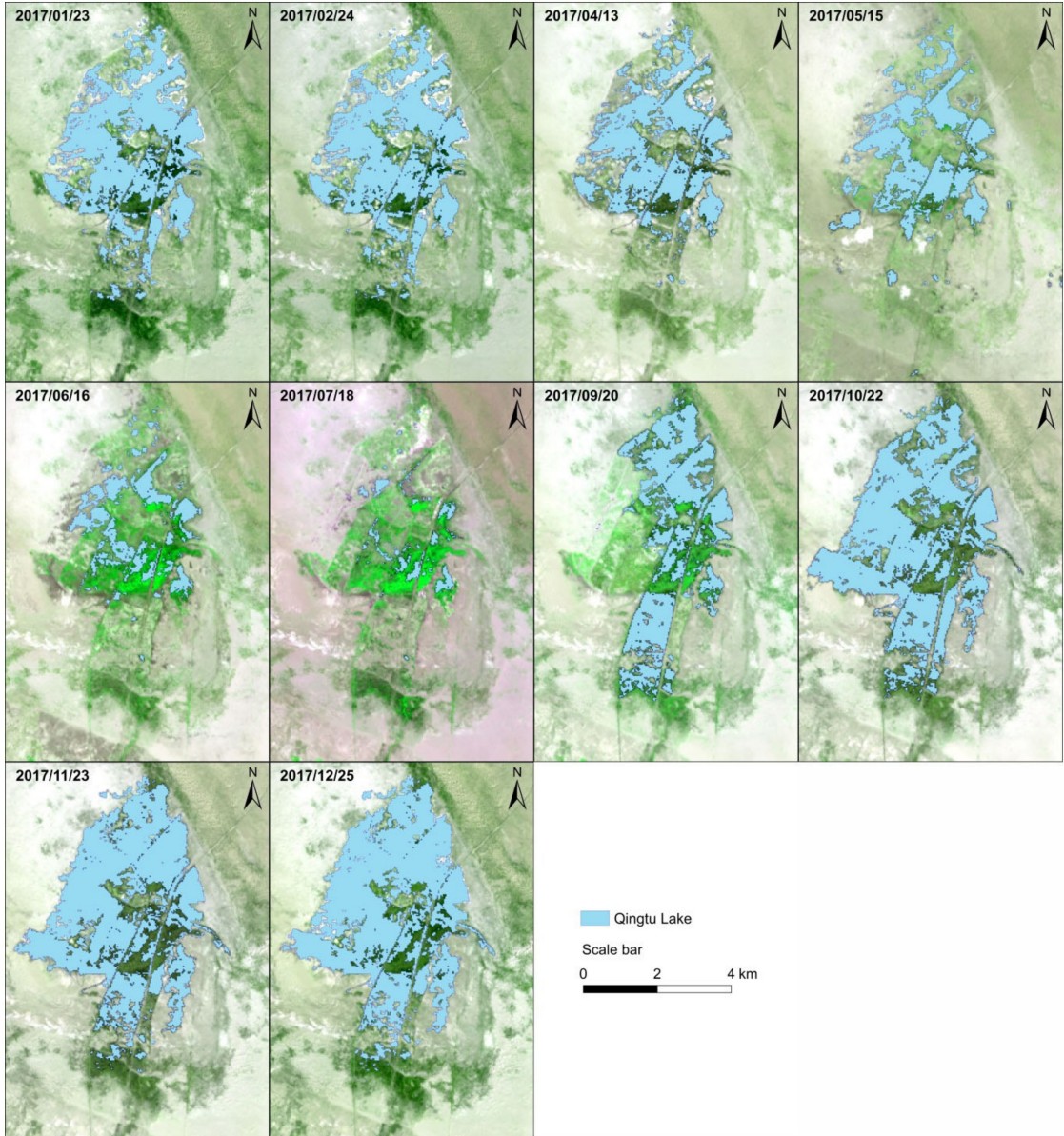

**Figure 4.** Spatial distribution of the Qingtu Lake water body from January to December 2017. The spatial trend of the lake water body distribution was similar in other years after 2010.

*4.2. Variation of Groundwater Level and Soil Water Saturation in Qingtu Lake Wetland Area*

4.2.1. Variation of the Groundwater Level

During the frozen period, the groundwater monitoring data loggers at locations G02, G08, and G10 were taken out from the wells in December 2018, and thus the data between December 2018 and May 2019 were missing. As shown in Figure 5, the groundwater level at each monitoring location showed a significant rise after 25 August, and then gradually stabilized. After November, the groundwater level at G05, which is located farther away from the lake center, first declined, and after January, the water level at G04 also showed a downward trend because of the groundwater recharge surface water. After March, the groundwater level at G04 and G05 rose obviously. It is speculated that this phenomenon was caused by the rise of the soil surface temperature, which made the frozen surface soil

water melt and recharge the groundwater. In winter, the groundwater at G03, G07, and G09 overflowed the surface, and the groundwater level remained relatively stable for a long time. It can also be seen that the direction of the regional groundwater flow is from G04 to the west and north, and from G9 to G08 and G10.

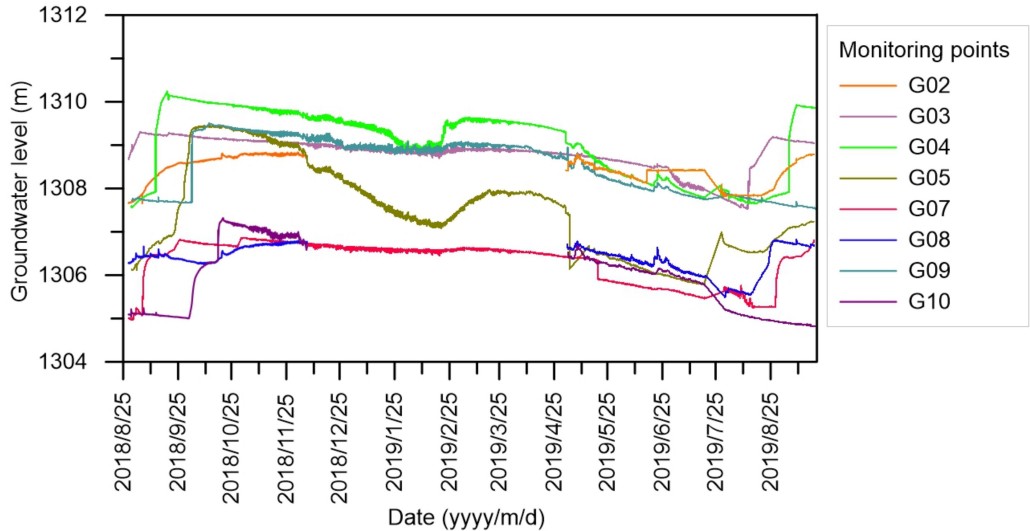

**Figure 5.** Groundwater-level hydrographs at various monitoring points in the Qingtu Lake wetland area from 2018 to 2019.

Long-term groundwater depth monitoring data at well W01 showed that the groundwater depth gradually decreased from July 2010 to December 2016 under the impact of the water transfer in autumn. The infiltrated surface water greatly increased the groundwater level. Within the same year, the overall groundwater depth fluctuated with greater depth from June to August (Figure 6).

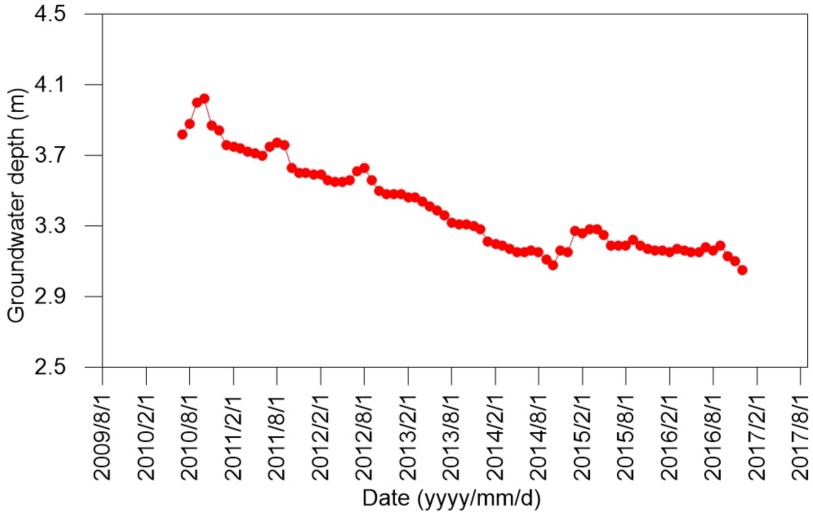

**Figure 6.** Long-term groundwater depth at location W01 near the Qingtu Lake wetland.

4.2.2. Variation in Soil Water Saturation

The water contents were converted to saturation according to Equation (5) [41]:

$$S_r = \frac{\theta}{n} \tag{5}$$

where $S_r$ represents the soil water saturation, $\theta$ is the soil water content, and $n$ is the porosity.

The monitored water contents were normalized by dividing by soil porosity to more clearly show the change in soil water content (Figure 7). The normalized data shown in Figure 7 refer to the soil water saturation. The V02 and V04 locations are close to the center of the lake, and the saturation at V02 and V04 was between 0.10 and 1. The V07, V08, and V09 locations are farther away, and the groundwater overflowed the surface after the water transfer each year. The saturation at V07 was between 0.388 and 1, the saturation at V08 was between 0.155 and 1, and the saturation at V09 was between 0.078 and 1. The V01, V05, V06, and V10 locations are farthest from the lake, and the groundwater could not submerge the surface; the saturation at V01 was between 0.02 and 1, the saturation at V05 was between 0 and 1, the saturation at V06 was between 0 and 0.471, and the saturation at V10 was between 0 and 0.791.

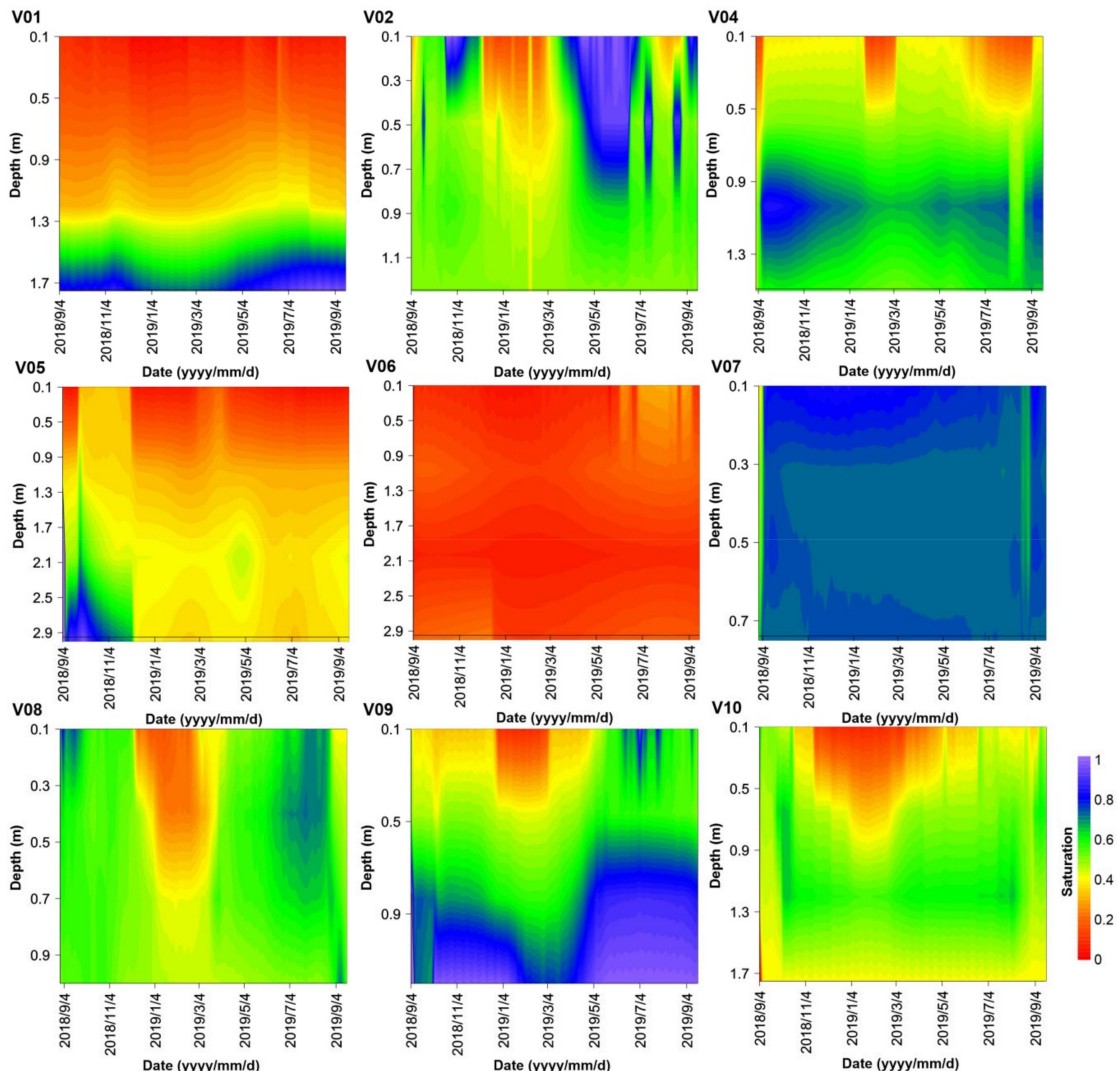

**Figure 7.** Variation of soil water saturation at various monitoring points.

As is shown in Figure 7, the variation of saturation at each location (except V06) was similar. The saturation of surface soil at each location was low, while the saturation of deep soil at each location was high. The saturation at the V02, V04, V07, V08, and V09 locations, which are closer to the lake, was low from January to March, and increased after March as the surface soil water melted. The conditions described are consistent with the variation characteristics of the groundwater level.

### 4.3. Variation in Vegetation Coverage around Qingtu Lake Wetland and Mingqin Basin

The vegetation classification of the entire Minqin Basin was calculated by referring to the vegetation coverage classification criteria in Table 1. Due to the influence of agricultural crops in the central basin [42], the interpreted vegetation classification with satellite data may not reflect the restoration of the vegetation cover area caused by water transfer. Therefore, this study interpreted the vegetation cover both in the entire Mingqin Basin and the natural vegetation within 10 km around Qingtu Lake. By comparing and analyzing the vegetation coverage degree within 10 km around Qingtu Lake and the entire Minqin Basin, we found that the characteristics of vegetation coverage were similar, and an obvious seasonal change trend was shown in the vegetation coverage areas for each category. Thus, we only showed the temporal and spatial vegetation coverage change within a 10 km area around Qingtu Lake in Figure 8. Generally, the lowest vegetation cover area occurred from the former November to the latter February, and the highest vegetation cover area occurred from June to September. However, there were some differences in vegetation cover area for each category. For example, the areas of bare rock, bare soil, and water, with the range of 0–10% vegetation coverage, varied little with the seasons, and the area was basically of the same order of magnitude. However, in the range of vegetation coverage above 30%, the vegetation coverage area varied greatly with the seasons, and the order of magnitude was not uniform. Sometimes, a season with an area of 0 may even occur. It is worth noting that, whether considering the entire Minqin Basin or a 10 km area around Qingtu Lake, the vegetation area with vegetation coverage above 70% was abnormal on 15 July 2010 and 18 July 2011. Compared with the data of the same period in other years, the magnitude difference was large. After analysis, it was found that the remote sensing images of these two days were from Landsat 4-5 TM and there were clouds, so it is speculated that the reason for this finding may be cloud interference in the remote sensing images. Meanwhile, the total vegetation coverage within a 10 km area around Qingtu Lake showed an obvious increasing trend since 2009 (Figure 8). The vegetation coverage in each category had increased, and the effect was most obvious in the vegetation coverage between 30 and 50%.

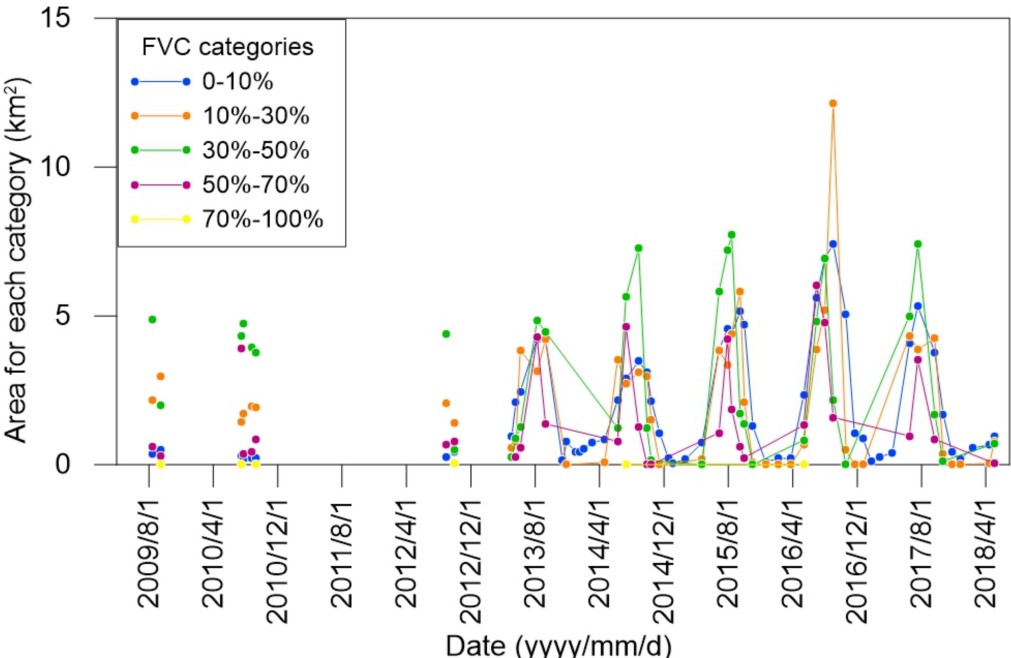

**Figure 8.** Variations over time of the vegetation coverage for different fractional vegetation coverage (FVC) categories in Qingtu Lake and its surrounding area within 10 kilometers. (Note: There is a lack of Landsat series satellite data from 2009 to 2013, among which there are only two data sets from August and September in 2009, four data sets from July, August and September in 2010, one data set from July, 2011, and two data sets from August and September in 2012).

Figure 9 shows the variation of land use from 1970 to 2017. The obvious changes were in the marshes and low coverage of grass of the natural oasis, which was influenced by the lake area. The area of low coverage grass of the natural oasis increased while the sand area decreased, which was consistent with the variation in vegetation coverage around the Qingtu Lake wetland (Figure 8).

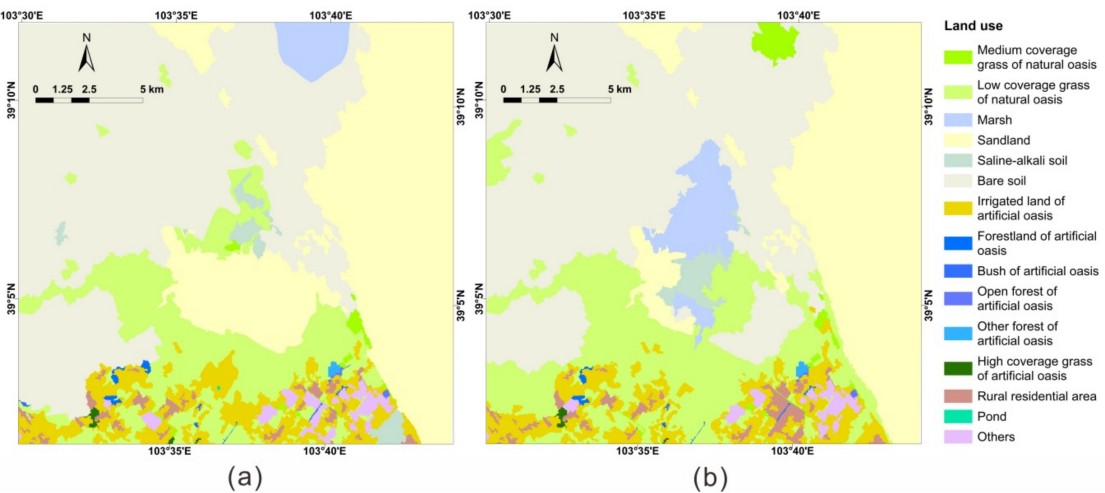

**Figure 9.** Variation of land use from 1970 (**a**) to 2017 (**b**).

### 4.4. Spatial Distribution of Vegetation Coverage in Qingtu Lake Wetland Area

The temporal change pattern of vegetation coverage is similar for different years, and thus we only show the spatial distribution of vegetation coverage from January to December 2017 in Figure 10. The areas with high FVC are mainly concentrated in locations close to Qingtu Lake and the south side of the study area, and the FVC of these areas varies greatly with the seasons. In the entire study area, the maximum FVC reached 18.6% from January to April. The maximum FVC reached 67.7% in June. The maximum FVC reached 68.7% in July. The vegetation coverage began to decline after September. In October, the FVC was low, and only sporadic areas had an FVC of more than 30%, while other areas had an FVC of less than 20%. After that, vegetation coverage continued to decline. The FVC, except at Qingtu Lake and the south side of the study area, was lower than 15% for a long time.

## 5. Discussion

Since September 2010, the water transfer to Qingtu Lake each year began at the end of August [40]. In order to explore the influence of this water transfer on the vegetation growth in the next year around Qingtu Lake, we analyzed the correlation between the water body area of Qingtu Lake in autumn and the maximum vegetation coverage in each FVC range in the following year, combining multiyear data. As shown in Table 2, the correlation coefficients varied among the vegetation areas, with different FVC ranges, and the area of the Qingtu Lake water body. The vegetation area with an FVC of 70 to 100% was small, making it difficult to show any correlation. The lake water body area and vegetation areas in other FVC ranges exhibited a positive correlation, suggesting that the more water transferred into Qingtu Lake in the previous autumn season, the larger the vegetation coverage in the following summer season. Compared with other FVC ranges, the vegetation coverage between 30 and 50% had a stronger correlation with the lake water body area. The correlation between the total vegetation coverage area and lake water body area was also strong, with an overall correlation coefficient of 0.839. As shown by the correlation coefficients, the transferred water had largest impact on the vegetation coverage, from 0 to 50% (Figures 3 and 8). Thus, we concluded that an increase in the vegetation coverage can contribute to the water transfer.

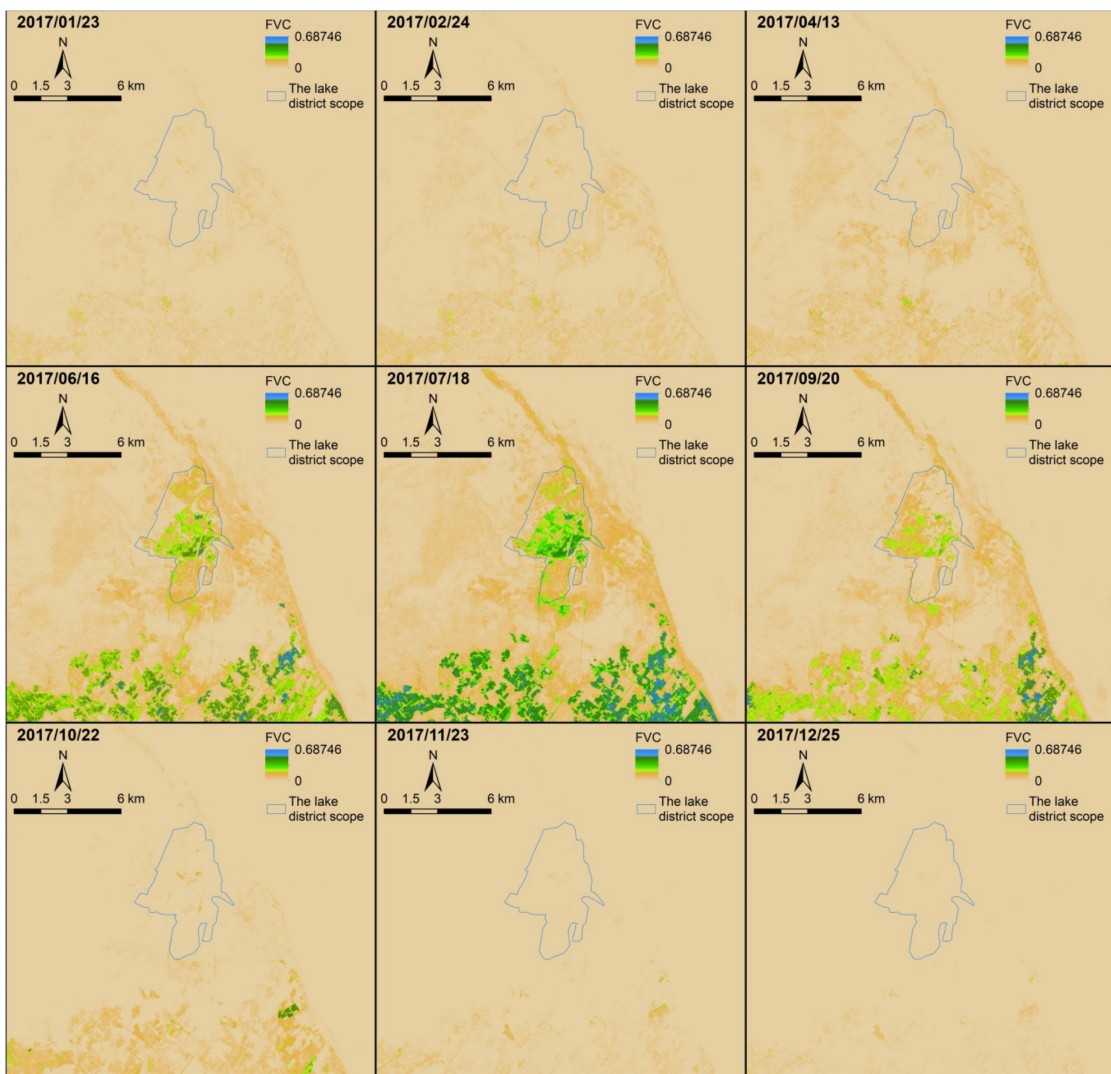

**Figure 10.** Variation of the spatial distribution of vegetation coverage near the Qingtu Lake wetland.

**Table 2.** Correlation Coefficients Between Different FVC Ranges and the Surface Area of the Water Body Formed by the Water Transfer.

| FVC | 0–10% | 10–30% | 30–50% | 50–70% | All |
|---|---|---|---|---|---|
| Correlation coefficient | 0.853 | 0.788 | 0.982 | 0.437 | 0.839 |

The rapid increase of the groundwater level in response to the water transfer suggested that the transfer surface water recharged groundwater (Figure 5). Then, the groundwater level slowly decreased, indicating the dissipation of infiltrated water. Following this, the thawing of frozen water also led to an increase of the groundwater level in spring. Long term, the groundwater-table depth has been slowly recovering since 2010, which could be ascribed to the infiltration from transferred surface water since 2010 (Figure 6). This process is similar to that which occurred in the Tarim River, where the water table near the riverbank was raised by 2–4 m during ecological water transfer during 2001 to 2006 [43]. Many studies reported that the decreased water table led to severe degradation of groundwater-dependent ecosystems in the arid and semiarid area [44], and thus, the increased water table in the study area has benefitted and improved the ecological environment for the vegetation system. A similar conclusion was reported in Heihe River Basin, northwest China [45], where the terminal East Juyan Lake has accumulated $6.19 \times 10^8 \, \text{m}^3$ water. The groundwater table rose by

an average 0.56 m downstream in the basin. This occurred after receiving ecological water transferred from upper and middle reaches from 2002 until about 2012, which led to an increase in areas of forest and grassland.

Responding to the surface water infiltration and groundwater level increase, increases in soil water content were observed at different monitoring locations, and higher water saturation began to be maintained from April to May (Figure 7). Thus, the soil water can be continuously supplied to the root zone and supports plant growth during spring [46]. In soil that undergoes seasonal freezing and thawing, reduced evaporation and seepage could be beneficial to conserve the soil water and maintain the high water content, and the thawing of frozen soil in spring also increased the water content (Figure 7) [47]. These soil water conditions in arid and semiarid areas are important for supporting the vegetation system. Thus, soil freeze–thaw processes are important for local ecohydrological processes, such as plant germination and growth in spring [48,49]. The transferred waters flowing over the ground surface could also infiltrate, to be either stored as soil water or recharged to groundwater, especially for the shallow distribution of clay soil [50]. The role of fine-textured soil in retaining water recharged by intermittent ecological water conveyances or prior floods as a lasting legacy to sustain riparian plant species over extended drought periods were also reported in the middle Heihe River, China [51] and the lower Rio Grande River, USA [52]. Through the interaction between surface water, groundwater and soil water, ecological water transfer in autumn increased the groundwater level and supported the relatively higher soil water content, providing essential water for vegetation during spring and summer in the following year.

## 6. Conclusions

To improve the degraded vegetation ecosystem in the arid areas of northwest China, caused by human activities and natural factors, artificial water delivery has become the main method of human intervention. Due to the contradiction between water use for agriculture and vegetation water demand in spring and summer, water transfer is generally carried out in autumn. However, the impact and mechanism of artificial water transfer on vegetation and wetland ecosystem restoration have not been thoroughly investigated, especially in northwest China. Taking the Qingtu Lake wetland system in the lower reach of Shiyang River in northwestern China as the study area, this study illustrated how water transfer affects the surface water, groundwater, and soil water interaction, and improves the vegetation ecosystem.

The surface area of Qingtu Lake expanded to the maximum in autumn and winter after water was transferred, and largely decreased the next summer of each year after 2010. The coverage of vegetation around Qingtu Lake area the next spring and summer also increased significantly each year from 2010. A positive correlation was found between the surface area of the lake body area in autumn and the vegetation coverage in each FVC interval the following summer, suggesting that water transfer improved the vegetation ecosystem in the study area. Further, we also explored the mechanism of how the water transfer in autumn, when the vegetation stops growing, improves the vegetation coverage in the following year. We found that the groundwater level increased after water transfer, and the soil water content increased and remained relatively high for the following months, which suggests that transferred water from upstream can be stored as groundwater or soil water in the subsurface through surface water. These water sources can provide the water for vegetation growth the next spring, or support the plants in the summer. Thus, ecological water transfer plays an important role in improving the vegetation ecosystem, even when the water is transferred in seasons when vegetation does not grow.

**Author Contributions:** Conceptualization, R.M.; methodology, S.Q.; validation, S.Q., R.M. and Z.S.; formal analysis, S.Q.; investigation, M.G., J.B., J.W., Z.W. and H.N.; resources, R.M. and Z.S.; data curation, M.G. and J.B.; writing—Original draft preparation, S.Q.; writing—Review and editing, R.M. and Z.S.; visualization, S.Q.; supervision, R.M.; project administration, R.M. and Z.S.; funding acquisition, R.M. and Z.S. All authors have read and agreed to the published version of the manuscript.

**Funding:** This research was financially supported by the National Key Research and Development Program of China (2017YFC0406105), National Natural Science Foundations of China (NSFC-41722208 and 41907177), and the Fundamental Research Funds for the Central Universities (CUGL180817).

**Acknowledgments:** We thank the Institute of Hydrogeology and Environmental Geology, Chinese Academy of Geosciences for their support for field trip and project coordination.

**Conflicts of Interest:** The authors declare no conflict of interest.

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
