# Peer review of "The Effect of Water Transfer during Non-growing Season on the Wetland Ecosystem via Surface and Groundwater Interactions in Arid Northwestern China"

_remotesensing, doi:10.3390/rs12162516_

Round 1
Reviewer 1 Report
In my opinion, this study is very interesting, but it needs major changes before publication. I cannot suggest publication at this stage. Some necessary changes are:
It is important to put the meaning of FVC (fractional vegetation coverage) where it is first mentioned.
Equation 4 and 5 have no reference.
In the “Study Area” section, it is important to specify the use of the lake water and if possible, to mention if it shows signs of contamination.
References are missing in the results and discussion section. It is necessary to reinforce the discussion section (mention some similar studies).
Regarding the figures:
The order of citing the figures is not correct (for example, figure 2 was found cited first and then figure 1).
Some figures are not cited in the text (for example, figure 6).
The letter size of the legends of the axes of the figures is not proportional.
It is not correct that the word “Legend” is presented in each figure:
- Figure 1. Change the word "Legend" of the images of the figure (put the appropriate text).
- Figure 4. Change the word "Legend" (put the appropriate text).
- Figure 5. Change the word "Legend" to "monitoring points".
- Figure 7. Change the word “Legend” at the range bar (put the appropriate text).
- Figure 8. Change the word "Legend" (put the appropriate text).
- Figure 9. Change the word "Legend" (put the appropriate text).
- Figure 10. Change the words "Legend" of all the images of the figure (put the appropriate text).
Figure 8 is not clear, it is not specified what the different colors mean, it only mentions percentages. This image needs to be improved.
Regarding Figures 4, 6, 9, and 10; Why it presents data from 2017 and not more current (2019)? It is important that the studies are recent.

Author Response
Response to Reviewer #1 Comments
All review comments are in black color while authors’ responses are in red color. And all revisions in the revised manuscript are in red color.
Reviewer #1: In my opinion, this study is very interesting, but it needs major changes before publication. I cannot suggest publication at this stage.
Response: We thank the reviewer for the positive comment and suggestions. These constructive feedbacks greatly enhanced this manuscript.
Comment 1: It is important to put the meaning of FVC (fractional vegetation coverage) where it is first mentioned.
Response: Thanks for the suggestion. The meaning of FVC has been put where it is first mentioned (Lines 25-26).
Comment 2: Equation 4 and 5 have no reference.
Response: Thanks for pointing this out. The references of equation 4 and 5 have been added in Lines 192 and 250, respectively.
Comment 3: In the “Study Area” section, it is important to specify the use of the lake water and if possible, to mention if it shows signs of contamination.
Response: There is no signs of contamination since there are only very few human activities in this wetland system and the maintenance of lake is mainly for the ecological purpose. This is described in the introduction section of the original manuscript “Qingtu Lake, located at the northern edge of Minqin Basin, is the terminal lake of the Shiyang River. It has important ecological significance in preventing the connection of the two big deserts”. We also specified the usage and the hydrochemistry characteristics of the lake water in the “Study Area” section (Lines 114-116) as shown in below.
“The water of Qingtu Lake is mainly used to maintain the ecological system balance around the wetland, especially for improving the vegetation ecosystem. The water in Qingtu Lake is weakly alkaline and Na–SO4•Cl type, with high salinity and TDS”
[22]Zhang, X.; Tao, H. Evolution and Conservation Policy of Wetland in Qingtuhu of the Minqin County. Gansu Sci. Tech. 2011, 27, 7-9.
[23]Zhang, Y.; Zhu, G.; Ma, H.; Yang, J.; Pan, H.; Guo, H.; Wan, Q.; Yong, L. Effects of Ecological Water Conveyance on the Hydrochemistry of a Terminal Lake in an Inland River: A Case Study of Qingtu Lake in the Shiyang River Basin. Water. 2019, 11, 1673.
Comment 4: References are missing in the results and discussion section. It is necessary to reinforce the discussion section (mention some similar studies).
Response: Thank you for the valuable comment. The references were added in the results and discussion sections and the discussion in the revised version was reinforced. We have compared our results with the ecological effect of Ecological Water transfer in Tarim River Basin and Heihe River Basin (Lines 351-360). We further discussed the ecological effect of water storage affected by soil freeze-thaw process and added citations (Yang et al., 2019; Wang et al., 2019; Yi et al., 2014) to support our finding that the thawing of frozen soil could increase the soil water content in spring (Lines 364-369). We added a citation (Chen et al., 2004) to prove the importance of soil water to support plant growth (Lines 363-364). In addition, we added three citations (Duan et al., 2016; Sun et al., 2016; Moore et al., 2016) to prove the role of fine-textured soil in retaining water recharged after intermittent ecological water transfer (Lines 372-375).
[40]Zhang, J. Problems and countermeasures of water resources schedule in hongyashan reservoir. J. Gansu Agric. Univ. 2015, 23-24.
[42]Xue, X.; Liao, J.; Hsing, Y.; Huang, C.; Liu, F. Policies, Land Use, and Water Resource Management in an Arid Oasis Ecosystem. Environ. Manage. 2015, 55, 1036-1051.
[43]Chen, Y.; Chen, Y.; Xu, C.; Ye, Z.; Li, Z.; Zhu, C.; Ma, X. Effects of ecological water conveyance on groundwater dynamics and riparian vegetation in the lower reaches of Tarim River, China. Hydrol. Process. 2010, 24, 170-177.
[44]Wang, Y.; Zheng, C.; Ma, R. Review: Safe and sustainable groundwater supply in China. Hydrogeol. J. 2018, 26, 1301-1324.
[45]Cheng, G.; Li, X.; Zhao, W.; Xu, Z.; Feng, Q.; Xiao, S.; Xiao, H. Integrated study of the water–ecosystem–economy in the Heihe River Basin. Natl. Sci. Rev. 2014, 1, 413-428.
[46]Chen, X.; Hu, Q. Groundwater influences on soil moisture and surface evaporation. J. Hydrol. 2004, 297, 285-300.
[47]Yang, K.; Wang, C. Water storage effect of soil freeze-thaw process and its impacts on soil hydro-thermal regime variations. Agric. For. Meteorol. 2019, 265, 280-294.
[48]Wang, T.; Li, P.; Li, Z.; Hou, J.; Xiao, L.; Ren, Z.; Xu, G.; Yu, K.; Su, Y. The effects of freeze-thaw process on soil water migration in dam and slope farmland on the Loess Plateau, China. Sci. Total Environ. 2019, 666, 721-730.
[49]Yi, J.; Zhao, Y.; Shao, M.a.; Zhang, J.; Cui, L.; Si, B. Soil freezing and thawing processes affected by the different landscapes in the middle reaches of Heihe River Basin, Gansu, China. J. Hydrol. 2014, 519, 1328-1338.
[50]Duan, L.; Huang, M.; Zhang, L. Use of a state-space approach to predict soil water storage at the hillslope scale on the Loess Plateau, China. Catena. 2016, 137, 563-571.
[51]Sun, Z.; Long, X.; Ma, R. Water uptake by saltcedar (Tamarix ramosissima) in a desert riparian forest: responses to intra-annual water table fluctuation. Hydrol. Process. 2016, 30, 1388-1402.
[52]Moore, G.; Li, F.; Kui, L.; West, J. Flood water legacy as a persistent source for riparian vegetation during prolonged drought: an isotopic study ofArundo donaxon the Rio Grande. Ecohydrology. 2016, 9, 909-917.
Comment 5: The order of citing the figures is not correct (for example, figure 2 was found cited first and then figure 1).
Response: Thanks for pointing this out. The order of citations of the figures has been revised in Line 106.
Comment 6: Some figures are not cited in the text (for example, figure 6).
Response: All the figures have been cited in the revised version as shown in Line 213, 244, 262.
Comment 7: The letter size of the legends of the axes of the figures is not proportional.
Response: The letter size of the legends and the axes of the figures have been revised to be proportional in the revised versions.
Comment 8: It is not correct that the word “Legend” is presented in each figure:
- Figure 1. Change the word "Legend" of the images of the figure (put the appropriate text).
- Figure 4. Change the word "Legend" (put the appropriate text).
- Figure 5. Change the word "Legend" to "monitoring points".
- Figure 7. Change the word “Legend” at the range bar (put the appropriate text).
- Figure 8. Change the word "Legend" (put the appropriate text).
- Figure 9. Change the word "Legend" (put the appropriate text).
- Figure 10. Change the words "Legend" of all the images of the figure (put the appropriate
text).
Response: All the words “Legend” in figures have been replaced with appropriate texts or as suggested words by reviewer. Please refer to the revised figures.
Comment 9: Figure 8 is not clear, it is not specified what the different colors mean, it only mentions percentages. This image needs to be improved.
Response: Thanks for the suggestion. The different colors indicate the range of FVC categories. We reivsed title of the vertical axe and the legend to make this more clear and the figure was improved in the revised version.
Comment 10: Regarding Figures 4, 6, 9, and 10; Why it presents data from 2017 and not more current (2019)? It is important that the studies are recent.
Response: We agree that it would be good to incorporate the most recent data. But some data are not accessible for the current stage. Since temporal trend of the Qingtu Lake water surface area in each year from 2009 to 2019 is similar to each other, there is no meaning to show all the changes for many years, and we only took the trend in 2017 as an example to illustrate the change of lake water area within the year, which would be similar to that in 2019. The similar reason is also for Figure 10 since the temporal change pattern of vegetation coverage is similar for different years. Until now, only the Landsat satellite data in January of 2019 can be downloaded. Thus, we don’t update the results for Figures 4, 9 and 10 in the revised version. We haven’t obtained the data for the long term monitoring well of groundwater (W01) after 2017 since these data were monitored by local water resource department. Thus, we didn’t present the most recent data after 2017 in Figure 6.

Reviewer 2 Report
Whilst not ground-breaking research, I am impressed with the quality and application of the science applied to understanding changes in vegetation following changed land-use (and river regulation) and subsequent attempts to improve the situation.
The application of remote sensing to record and describe the temporal changes across the landscape are well devised and executed and clearly presented.
I was left thinking the more could be made of the data and perhaps greater rigour in the selection of customised indices (especially as the satellite generating the data for the NDVI index has recently been shown to have drifted and requires re-calibration), but for the application the results are logical and justify the use of the bands selected.
I only have one minor edit - please expand the P. australis reference to Phragmites australis in line with the other vegetation species. Otherwise, the text is fine and in good English.
I think this is a good addition to the literature on vegetation response to human intervention, both indirectly due to diversions and directly through managed release.
I recommend publication.
Author Response
Response to Reviewer #2 Comments
All review comments are in black color while authors’ responses are in red color. And all revisions in the revised manuscript are in red color.
Reviewer #2: Whilst not ground-breaking research, I am impressed with the quality and application of the science applied to understanding changes in vegetation following changed land-use (and river regulation) and subsequent attempts to improve the situation.
The application of remote sensing to record and describe the temporal changes across the landscape are well devised and executed and clearly presented.
I was left thinking the more could be made of the data and perhaps greater rigour in the selection of customised indices (especially as the satellite generating the data for the NDVI index has recently been shown to have drifted and requires re-calibration), but for the application the results are logical and justify the use of the bands selected.
I only have one minor edit - please expand the P. australis reference to Phragmites australis in line with the other vegetation species. Otherwise, the text is fine and in good English.
I think this is a good addition to the literature on vegetation response to human intervention, both indirectly due to diversions and directly through managed release.
I recommend publication.
Response: We are grateful to Reviewer 2 for the positive comments.
The word “P. australis” has been replaced by “Phragmites australis” in Line 109 as suggested.

Round 2
Reviewer 1 Report
In my opinion, this study presents an improved form.
I can suggest its publication.